# Determination of the Prevalence of Microsatellite Instability, *BRAF* and *KRAS*/*NRAS* Mutation Status in Patients with Colorectal Cancer in Slovakia

**DOI:** 10.3390/cancers16061128

**Published:** 2024-03-12

**Authors:** Tomas Rendek, Rami Saade, Ondrej Pos, Georgina Kolnikova, Monika Urbanova, Jaroslav Budis, Luboslav Mihok, Miroslav Tomas, Tomas Szemes, Vanda Repiska

**Affiliations:** 1Institute of Medical Biology, Genetics and Clinical Genetics, Faculty of Medicine, Comenius University, 811 08 Bratislava, Slovakia; vanda.repiska@fmed.uniba.sk; 22nd Department of Gynaecology and Obstetrics, Faculty of Medicine, Comenius University, 811 08 Bratislava, Slovakia; rami.saade@fmed.uniba.sk; 3Geneton Ltd., 841 04 Bratislava, Slovakia; ondrej.pos@uniba.sk (O.P.); tomas.szemes@geneton.sk (T.S.); 4Science Park, Comenius University, 841 04 Bratislava, Slovakia; 5Department of Pathological Anatomy, National Cancer Institute, 833 10 Bratislava, Slovakia; georgina.kolnikova@nou.sk (G.K.);; 6Department of Medical Genetics, National Cancer Institute, 833 10 Bratislava, Slovakia; 7National Cancer Institute, Surgical Oncology Clinic of Slovak Medical University, 833 10 Bratislava, Slovakia

**Keywords:** microsatellite instability, colorectal cancer, BRAF, KRAS, NRAS mutations

## Abstract

**Simple Summary:**

Identifying the mutation status of driver genes in patients with colorectal cancer is a standard clinical practice in oncology nowadays. Despite this, molecular biomarker data for CRC patients on a national level in Slovakia are limited. Our study analyzed 83 CRC patient tumor tissues from the National Cancer Institute (NCI) database, examining microsatellite instability (MSI), BRAF, KRAS/NRAS mutations, and neoplastic cell percentage. Results revealed 4 MSI-high samples, 39 KRAS/NRAS mutations, and 5 BRAF p.V600E mutations, with one case exhibiting all three markers. We explored relationships between biomarkers, their coexistence, and demographic factors. This research addresses the scarcity of molecular data in Slovakia’s CRC landscape, contributing valuable insights into biomarker prevalence and interactions within the population.

**Abstract:**

Slovakia has one of the highest rates of colorectal cancer among the developed countries, ranking as the second highest in the incidence of this disease for men worldwide. Despite the significant burden on both quality of life and the healthcare system this disease imposes, data on molecular analysis of biomarkers in CRC-diagnosed patients is scarce. In our study, we analyzed confirmed CRC patients from the database of the National Cancer Institute (NCI) and evaluated the presence of 4 biomarkers in tumor tissues. Altogether, 83 FFPE tumor tissues from CRC patients listed in the NCI database were analyzed for microsatellite instability status, presence of BRAF and KRAS/NRAS mutations, and neoplastic cell percentage in tissue samples. We identified 4 MSI-high samples, 39 KRAS/NRAS mutations, and 5 BRAF p.V600E mutations, with one case of coexistence of all three markers in a single tumor sample. We also evaluated possible relationships between biomarkers, their coexistence, and the age and sex of the studied population.

## 1. Introduction

Slovakia is one of the countries with the highest incidence of colorectal cancer (CRC), mounting at 43.9 cases per 100,000 residents, which accounts for 15.9% of all new cancer cases, based on the 2020 WHO International Agency for Research on Cancer [1,2]. Due to its high socioeconomic burden, studying individual cases of CRC in the Slovak Republic is imperative. In recent years, local research groups have been assessing the effectiveness of national screening programs for CRC, utilizing fecal immunochemical tests (FIT) on simulated population models. [3] Moreover, local researchers aimed to repurpose local prenatal screening data to identify genetic variants associated with colorectal cancer, focusing on genes associated with Lynch syndrome as the most common germline mutations associated with the development of CRC. [4] A recently published joinpoint analysis of local CRC trends showed only slight improvement in reduced mortality (0.9%) in the follow-up period from 2001 to 2018 compared to the previous period from 1971 to 2001, limited to the age group from 29 to 45 years old [5]. Due to the lack of improvement in CRC prevalence, the identification of possible driver mechanisms can help to elucidate the high incidence as well as mortality of CRC in Slovakia, as the underlying cause of this problem is still unclear, with a limited number of regional studies being conducted [4]. 

CRC can arise by either one or a combination of three mechanisms: chromosomal instability, microsatellite instability (MSI), or CpG island methylator phenotype (CIMP) [6]. Chromosomal instability is considered to be the most prevalent mechanism, responsible for 60–70% of sporadic CRC cases, where chromosomal imbalance resulting in aneuploidy and loss of heterozygosity in somatic cells leads to cancer development [7]. In CIMP tumors, the methylation of CpG islands within the promoter region leads to the transcriptional silencing of genes that play a role in tumor suppression or are involved in cell cycle regulation [7,8]. The exact definition of CIMP tumors varies across the literature, with methylation status of 5 loci: hMLH1, p16, MINT1, MINT2, and MINT31 being usually investigated [7,8]. MSI is a phenomenon followed by the deactivation of one or more mismatch repair (MMR) genes due to their mutation or epimutation, which maintains genome integrity by repairing errors emerging from DNA replication [9]. A substantial portion of CRCs, especially in early-onset cases, can arise due to hereditary disorders characterized by germline mutations in associated genes. The underlying mechanisms of early-onset CRC cases have been extensively studied in connection with hereditary syndromes, mainly familial adenomatous polyposis and hereditary nonpolyposis colorectal cancer, also known as Lynch syndrome (LS) [10]. Microsatellites also called short repetitive repeats, consist of repeating sequences of 1–6 nucleotides. The characteristics of their distribution are variable from 15 to 65 tandem nucleotide repeats of small satellite DNA located mainly at the ends of chromosomes. Microsatellites are widely distributed and can often also be found in the vicinity of exons or introns [11]. The mechanism of microsatellite formation can generally be attributed to errors of DNA polymerase during replication, where redundant repeats of a certain sequence may be formed, or, conversely, a certain number of repeats may be lost compared to the template strand. Normally, such errors are repaired by the MMR mechanism, but in case of its failure, such new variants are preserved, contributing to the accumulation of mutations in the affected cells and significantly impacting their transformation and subsequent progression into cancer cells [12]. Depending on the degree and frequency of MSI, it can be divided into three categories: high microsatellite instability (MSI-H), low microsatellite instability (MSI-L), and microsatellite stability (MSS) [13]. In recent studies, MSI-L and MSS are classified into the same category as MSS [14]. MMR mechanism, which is responsible for sustaining genome stability [14,15], consists of the following genes: MLH1, MSH2, MSH6, and PMS2, which have proven to hold a high clinical significance for MSI and genetic predisposition for CRC and PMS1 and MLH3 with less clear significance in CRC development [16]. 

Proto-oncogene BRAF is a potent modulator of the signaling pathway known as Ras/Raf/MEK/ERK [17]. BRAF V600E has been studied as a driver mutation associated with the development of CRC [18]. Alterations in the BRAF gene seem to occur early in CIMP tumors. The specific BRAF V600E mutation shows a significant association with MLH1 promoter hypermethylation and is present in approximately 20.3% of cases involving CRC [19]. BRAF mutation is believed to be responsible for the failure of anti-EGFR (epidermal growth factor receptor) therapy in approximately 10–15% of cases [17] and is also valuable as a negative prognostic factor, especially in metastatic and MMR deficient CRCs [20]. RAS proteins function as compact GTPases, serving as molecular switches for relaying signals from activated receptors. They accomplish this by transitioning between an inactive state bound to GDP and an active state bound to GTP. In its GTP-bound state, RAS can interact with and activate various downstream effector proteins, leading to a variety of cellular responses such as cell proliferation, survival, differentiation, and potential neoplastic transformation [21]. Approximately 30% of human cancers, comprising both solid tumors and hematologic malignancies, are connected to alterations in RAS genes. Interestingly, mutations in distinct RAS isoforms show a predisposition to different types of cancer affecting various organs [22]. KRAS mutation status is imperative for identifying patients who would benefit from anti-EGFR therapy, making it one of the prime examples of predictive biomarkers for personalized cancer treatment [23]. The prognostic value of KRAS mutations is more unclear, with studies suggesting worse outcomes for patients carrying mutations in codons 12 and 61 [24]. Mutations of NRAS have a much lower incidence than KRAS or BRAF, as they are identified in approximately 2–4% of CRCs. Thus, the reason why the roles of NRAS mutations as prognostic and predictive markers in metastatic CRCs were much less investigated. A meta-analysis of CRC outcomes concluded that patients carrying NRAS mutations have an overall worse progression-free survival rate [25].

Accurate identification of abnormalities in the KRAS, NRAS, and BRAF genes holds significant importance for the appropriate assessment of CRC treatment by anti-EGFR monoclonal antibodies [26]. The concomitant KRAS and BRAF mutations occur rarely in CRCs. It is also uncommon practice to test already KRAS-positive patients for BRAF mutation; hence, this occurrence is usually identified only in clinical studies [27]. Even in larger studies, the existence of KRAS and BRAF appear to be mutually exclusive [28]. In our retrospective study, we aim to assess the prevalence of the mentioned driver mutations in CRC patients, along with other clinical factors, and compare their prevalence with global data. This study also aims to support any follow-up research on the high morbidity and mortality of CRC patients in Slovakia. 

## 2. Materials and Methods

### 2.1. Ethical Approval

Here, we provide a retrospective cohort study, analyzing 83 patient records collected by the Department of Pathology of the National Cancer Institute in Bratislava, Slovak Republic, between 2020 and 2022—all included patients undergoing routine diagnostic testing signed informed consent. The study was approved by the Ethics Committee of the National Oncology Institute in Bratislava, protocol code: PATOL-01.

### 2.2. Cohort Specification

The following criteria were used to specify the cohort of patients for whom genetic testing was requested: (I) a patient with histopathological confirmation of CRC diagnosis categorized by the International Classification of Diseases 10th Revision (ICD-10) as summarized in Table 1; (II) a failure of MMR protein expression proven by immunohistochemical (IHC) analysis, or if the staining result was inconclusive; (III) patient undergoes genetic testing for MSI status; (IV) patient undergoes BRAF, NRAS, and KRAS genetic examination. 

### 2.3. Sample Preparation

The tumor samples were collected either as part of a tissue biopsy examination or tissue resection during a curative surgical procedure. Tumor tissue was processed for formalin fixation and paraffin embedding (FFPE) at the Department of Pathological Anatomy of NCI. The indicated CRC samples were sent for examination of MSI status as well as the genetic examination of BRAF and KRAS/NRAS mutations at the Department of Medical Genetics, NCI. The DNA isolation from FFPE samples was performed according to standardized procedures in clinical genetics laboratories. 

### 2.4. Detection of Microsatellite Instability

For identification of microsatellite unstable tumor tissues, 5 mononucleotide microsatellite markers were used: NR-21, BAT-26, BAT-25, NR-24, and MONO-27, followed by fragmentation analysis on ABI PRISM 3130 genetic analyzer. Alternatively, MSI status was investigated with Idylla™ MSI Assay on the Biocartis Idylla™ System platform. Both assays are standard methods for MSI detection at the Department of Medical Genetics, NCI. 

### 2.5. Identification of BRAF and KRAS/NRAS Mutations

Evidence of the BRAF gene’s most prevalent mutations, V600E and V600K, respectively, were obtained using allele-specific PCR with the Cobas Z 480 PCR system (Roche Molecular Diagnostics, Pleasanton, CA, USA). The status of RAS genes was investigated on the Roche LightCycler 480 platform (Roche Molecular Diagnostics) using the RAS kit Mutation Screening Panel CE-IVD (Entrogen, Woodland Hills, CA, USA), targeting KRAS and NRAS genes (codons 12/13, 59/61, 117/146).

## 3. Results

In total, we searched through approximately 3500 requests for genetic testing for the years 2020–2022, of which 83 patients met our criteria. The detailed profile of each of the enrolled patients in our study is provided in Appendix A. We summarize the results of the MSI status, *BRAF*, *KRAS*, and *NRAS* genes and the results of the IHC examination of MMR protein expression in tumor tissue in Table 2. 

Overall, four patients with MSI-H status of their tumor tissue samples were identified, making up 4.8% of cases from the studied group. Identified MSI-H samples were investigated for MMR gene expression status through IHC. The results of the MMR expression status are summarized in Table 3.

A better visualization of all patient parameters, along with mutation status, is provided in the plot (Figure 1).

The age of patients in our study varied from 26 to 86 years, with an average of 62.5 years. The median age for males and females was 61.81 and 63.24 years, respectively. The male population was slightly larger, accounting for 55.42% of the studied group. The distribution of patients into age groups according to their gender is visualized in Figure 2.

## 4. Discussion

We searched the NCI database of genetic testing applications of CRC-confirmed patients between 2020 and 2022. It is worth mentioning that based on our criteria described in the chapter Methods, we were able to select only 83 patients from approximately 3500 applications for MSI testing. The possible reasons for this might be the fact that in clinical practice, the occurrence of *RAS* and *BRAF* mutations are often seen as mutually exclusive; thus, after identification of one mutated driver gene, the mutation status of the latter will not be investigated, despite having a clinical significance [29]. This also posed the biggest limitation for our study, narrowing down our cohort to only 2.37% of patients enrolled for MSI testing. The aim of this research was to identify patients who were investigated for MSI, *BRAF,* and *KRAS*/*NRAS* mutational status. From the available results of the individual examinations, we summarized in Appendix A the age and gender of the patients, the diagnosis according to ICD-10, the neoplastic cell percentage, the mutational status of *KRAS*, *NRAS*, and *BRAF* genes, the MSI status, and the expression status of MMR proteins based on IHC staining in MSI-H patients. However, these data may be useful in possible future investigations. Gender distribution in our study cannot be correlated with the gender distribution of CRC incidence in Slovakia due to applied selection criteria, which do not reflect the total number of newly diagnosed cases by gender. Based on GLOBOCAN data, an almost 2-fold higher incidence in male patients than females was to be expected [30]. The age of patients in our cohort ranged from 26 to 86 years, with a median age of 62.5 years and a median age for males and females of 61.81 and 63.24 years, respectively. Up to 66.3% of the patients were older than 60, which aligns with literature describing the late onset of sporadic CRC in developed countries [31]. 

The importance of screening for genes encoding RAS proteins, specifically *KRAS* and *NRAS*, lies primarily in the selection of appropriate chemotherapy [32]. Mutations in the *KRAS* gene occur independently of MMR status; however, a higher incidence of *KRAS* mutations has been linked to mutations in the *MSH2* and *MSH6* genes, where it is implicated in the more rapid progression of carcinogenesis [33]. A direct association between hereditary forms of colorectal cancer and *RAS* gene mutations has not been demonstrated, but simultaneous mutation of both groups of genes is of major prognostic and therapeutic importance. In our patient cohort, the mutation in the *RAS* gene was detected in 39 patients (47%), from which *KRAS* mutation was detected in 33 (39.8%) patients, and *NRAS* gene mutation in 6 (7.2%) patients. Our results are in accordance with the global prevalence of RAS family mutation prevalence in CRC, as noted in clinical trials meta-analysis by Sorich [34] and an analysis of U.S. population prevalence [35]; both prevalences were approximately 50%. Detection of the *BRAF* gene mutation, specifically V600E, is a useful marker to differentiate sporadic and hereditary forms of MMR deficiency because of its close association with MLH1 promoter hypermethylation, which is responsible for approximately 70% of sporadic forms of MMR-deficient CRC. The *BRAF* V600E mutation was detected in 9 (4.6%) patients; thus, the hereditary background of the disease (LS) can almost certainly be ruled out in these patients [32]. *BRAF* mutation prevalence in our cohort was similar to the prevalence of *RAS* family mutations in accordance with global prevalence. An examination of MSI status is a crucial factor that indicates the state of function of the MMR machinery in the cell. However, MSI examination alone does not serve to differentiate hereditary from sporadic forms of MMR deficiency, which has predictive and prognostic value for the patient’s family members in addition to its prognostic value for the patient. However, in the case of MSS tumors, the LS can be ruled out [14]. According to the literature, microsatellite instability is present in approximately 15% of colorectal cancers, while the majority of cases are caused by somatic MLH1 promoter hypermethylation compared to MSI caused by LS [36]. In the case of positive MMR protein expression, according to the literature, there is a low probability of MSI-H in tumor tissue. However, the sensitivity and specificity of IHC analysis of MMR protein expression are greatly influenced by the quantity and quality of the material as well as the pathologist’s experience. The quality of the sample obtained for tumor testing can be ascertained by the percentage of neoplastic cells (NCP). Low NCP (patients 12, 61, 69) might suggest that the sample contains a major proportion of healthy tissue instead of the tumor the biopsy was targeted for, which can lead to a false negative test [37]. The major limitation in methods lies in the IHC analysis of MMR protein expression. In certain cases, there may be a discordance between the results of the IHC analysis of MMR proteins and the PCR examination of MSI status [38]. In such a case, both examinations should be repeated if the quantity and quality of the archived material permit. In our case, we were able to detect a loss of expression of at least one of the MMR proteins in all patients with MSI-H status. We summarize the results of the IHC examination of MMR proteins and MSI status in Table 2. Only four of these patients were detected to have MSI-H status with a confirmed knockout of expression of MMR proteins, and all of them were older than 60 years of age. In addition, 2 of them (50%) had a mutation in the *BRAF* gene. Three patients were aged less than 60 years, specifically 33, 44, and 45 years, but the former had a positive MMR protein expression, and the remaining two patients were missing the results of *BRAF* gene testing and MMR protein expression analysis. According to the literature, *KRAS* mutations occur in 35% to 45% of CRC, mostly in codon 12 (80%), c.35 G > A (G12D), and c.35 G > T (G12V) transversions, representing 32.5% of *KRAS* mutations [39]. In our research, 39.75% of patients were positive for *KRAS* mutation, which is similar to the percentage reported in the literature. 

The co-occurrence of *BRAF* and *KRAS* mutation in patient number 35 was a unique phenomenon, emerging in about 0.001% of CRC cases [27], and is scarcely discussed in the available literature, mostly through case reports of individual patients. For example, such mutation co-occurrence was discovered even within one cancer cell during a single-cell sequencing study conducted by Gularte-Merida et al., indicating that dual-driver mutations are present in a rare subgroup of CRCs [40]. This co-occurrence also supports the hypothesis of two clonal origins of CRC in this particular patient, as was the case in another case study [27]. The co-occurrence of BRAF and KRAS mutation also has predictive value, apart from prognostic. Patients with advanced CRC with wild-type RAS are eligible for cetuximab or panitumumab, whereas patients with BRAF p.V600E–mutant CRC are eligible for combination therapy with encorafenib and cetuximab. [41]. In current discussions, there is a suggestion that addressing multiple RAS mutations in colorectal cancer (CRC) may require treating them as distinct primary conditions, as they might evolve separately in parallel evolutionary pathways and be present in lower copy numbers even before the administration of treatment, as was shown in primary melanoma study [42]. In our study, KRAS and BRAF co-occurrence was accompanied by MSI-H status. This is confirmed by research in other countries, such as the team of Rodrigo Gularte-Mérida, where the co-occurrence of KRAS and BRAF mutation is reported in MSI-H tumors exclusively [40]. The presence of BRAF mutation in colorectal cancer is by itself a negative prognostic factor, as therapeutic strategies utilizing medications aimed at BRAF-V600E in CRC have shown comparatively lesser efficacy than, for example, in BRAF mutant melanoma. Specifically, BRAF-V600E inhibitor monotherapy has yielded an overall response rate of around 5% [43]. In addition, KRAS mutation has additional adverse effects and is reported to significantly increase disease progress in such patients [43,44,45,46]. On the other hand, prognostic and predictive value in MS status is less clear. In general, patients with microsatellite unstable tumors in stage II colorectal cancer have an overall more favorable prognosis [47]. In later stages, the overall prognostic value of MS status was less clear, and MS-instable patients did not receive any benefit from adjuvant chemotherapy by 5-Fluorouracyl in 5-year survival compared to patients with MS-stable tumors [48]. There are published meta-analyses, notably by Wang et al., which are also disputing the prognostic value of MSI status for advanced stages of colorectal cancer [49] 

## 5. Conclusions

MSI testing plays a crucial role in categorizing CRC tumors into MSI-H and microsatellite stable types. Among all CRCs, MSI-H or MMR-deficient tumors have displayed the most favorable prognosis, making MSI testing a valuable prognostic marker. Additionally, it aids in identifying LS among familial CRC patients. A diagnostic mutation in the BRAF gene (V600E) can potentially distinguish sporadic from hereditary CRCs, as it aligns with sporadic cases. While some earlier studies highlighted the predictive role of MSI testing in chemotherapy, recent research has presented conflicting results, leading many authors to refrain from recommending it as a routine examination for assessing therapeutic response. 

The prevalence of main driver mutations of CRC in our study was in alignment with globally published literature; however, the size of the studied cohort was limited due to the small number of patients undergoing simultaneous testing for *BRAF*, *KRAS* mutation status, as well as microsatellite stability status. Nonetheless, identifying MSI along with the most common driver mutations and their co-occurrence provides a better picture of the molecular background of the high incidence of CRC in the Slovak population, strongly encouraging prospective investigation. Further research should aim for a broader cohort of patients, extended by investigation of family history, screening for Lynch syndrome-associated mutations according to Bethesda testing guidelines, and focus on therapeutic implications for the patients enrolled in future studies due to the rising burden of this disease in Slovakia and worldwide.

## Figures and Tables

**Figure 1 cancers-16-01128-f001:**
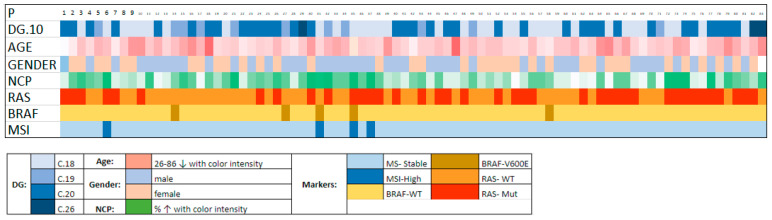
Plot of patients (P) enrolled in the study, visualizing diagnosis, age, gender, neoplastic cells percentage (NCP) in a tissue sample, mutation status of RAS, BRAF, and MSI status. The color representation of each parameter is explained in the legend below the plot.

**Figure 2 cancers-16-01128-f002:**
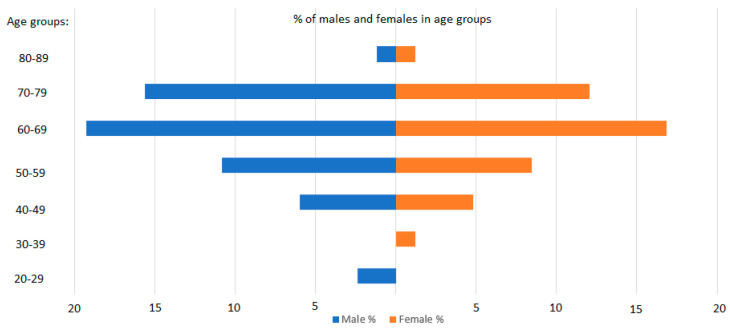
Percentage distribution of patients according to sex and age, clustered into age groups.

**Table 1 cancers-16-01128-t001:** List of CRC subtypes according to ICD-10 with quantities of patients involved in our research.

Diagnosis	ICD-10 Code	No. of Patients
Malignant neoplasm of colon	C18	3
Malignant neoplasm of cecum	C18.0	6
Malignant neoplasm of ascending colon	C18.2	4
Malignant neoplasm of splenic flexure	C18.5	3
Malignant neoplasm of sigmoid colon	C18.7	13
Malignant neoplasm of overlapping sites of colon	C18.8	1
Malignant neoplasm of colon, unspecified	C18.9	4
Malignant neoplasm of rectosigmoid junction	C19	11
Malignant neoplasm of rectum	C20	29
Malignant neoplasm of anus, unspecified	C21.0	6
Malignant neoplasm of intestinal tract, part unspecified	C26.0	2
Malignant neoplasm of ill-defined sites within the digestive system	C26.9	1

**Table 2 cancers-16-01128-t002:** Summary of results of MSI status and mutation status of BRAF, KRAS, and NRAS genes with the percentage of affected patients.

	MSI-H	*BRAF*	*KRAS*/*NRAS*
Positive	4 (4.8%)	5 (6.0%)	*KRAS*: 33 (39.8%)
			*NRAS*: 6 (7.2%)
Negative	79 (95.2%)	78 (94.0%)	44 (53.0%)

**Table 3 cancers-16-01128-t003:** Results of genetic testing and immunohistochemical detection of MMR protein expression in patients with MSI-H tumors.

P	ICD-10	Age	Gender	NCP	*RAS*	*BRAF*	MSI	MMR
6	C18.2	62	F	100	*KRAS*: c.38G > A, p.G13D	N	MSI-H	MLH1-, PMS2-
31	C18	74	M	80	N	p.V600E	MSI-H	MLH1-, PMS2-
35	C18.2	79	F	70	*KRAS*: c.35G > A, p.G12D	p.V600E	MSI-H	MLH1-, MSH2-, PMS2-
37	C18.2	67	M	90	*KRAS*: c.35G > A, p.Gly12Asp	N	MSI-H	MLH1-

P—patient number; NCP—neoplastic cells percentage in the examined sample; *RAS*—mutational status; *BRAF*—mutational status; MMR—expression of MMR proteins by IHC; N—no mutation present; symbol “-“ after a specific name of MMR protein indicates the loss of its expression.

## Data Availability

The original contributions presented in the study are included in the Appendix A. Further inquiries can be directed to the corresponding author.

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
