# Peer review of "Determination of the Prevalence of Microsatellite Instability, BRAF and KRAS/NRAS Mutation Status in Patients with Colorectal Cancer in Slovakia"

_cancers, 2024, doi:10.3390/cancers16061128_

Round 1

Reviewer 1 Report

Comments and Suggestions for Authors

The authors collated information from the NCI database on microsatellite instability, BRAF and KRAS/NRAS mutation status in tumor tissue of Slovak colorectal cancer patients between 2020 and 2022. The above information of 83 patients was listed, which supplemented molecular biomarker data of CRC patients on a national level in Slovakia to some extent. However, we believe that both the richness and number of cases in the study are insufficient, leading to a limited guidance. They should be expanded and supplemented comprehensively. Here are some specific problems and suggestions for rectification.

Major comments:

1.Meticulously analyzing over 3500 data points to identify 83 cases with comprehensive information was an arduous task, and we commend the authors for their diligent efforts. However, the discrepancy lies in the sex ratio of selected cases, which does not align with prevailing epidemiological trends in Slovakia. Furthermore, the MMR gene deletion profile in MSI-H patients is not sufficiently comprehensive. Therefore, we strongly advocate for the inclusion of a larger sample size and integration of additional record years by the authors to enhance the credibility and acceptance of their findings.

2.We recommend incorporating the patient's clinical or pathological TNM stage and the tumor tissue's pathological classification to enhance the provision of more precise prognostic guidance.

3.The mere enumeration of the data indicates a lack of research value in the paper, and we believe it would be more advantageous to utilize the acquired data for pertinent statistical or bioinformatics analyses.

4.

5.The incorporation of PD-L1 scores and even Her-2 expression is recommended for enhancing the clinical treatment guidance value in MSI-H patients.

6.The p.V.600E mutation has been identified as a robust indicator for differentiating between sporadic and familial MMR deficiency, as stated in [page 7 line 212]. Consequently, it is advisable to gather supplementary diagnostic information regarding Lynch syndrome in patients exhibiting both the BRAF P.V.600E mutation and MSI-H status (patients 31 and 35).

7.The deletion pattern of the MMR gene in patient 35, as observed in Table 3, demonstrates an exceptionally low incidence rate within MSI colorectal cancer. Therefore, it is crucial to validate and reaffirm the IHC profile of the MMR gene.

Minor comments:

1.The legend and text in Figure 1 should consistently employ the appropriate terms "sex" or "gender".

2.The current legend of Age in Figure 1 only includes one red color scheme, which fails to accurately represent the age categories depicted in the picture through color variations. It is recommended to incorporate a color scheme that precisely reflects the different age groups portrayed in the image.

3.The title of reference [26] on page 11, line 337 lacks completeness.

Comments on the Quality of English Language

The author's English expression is consistently smooth and polished overall, with only a few minor suggestions for improvement as mentioned in Minor comments, without any additional recommendations.

Author Response

Dear reviewer, on behalf of all authors we would like to thank you for your involvement in reviewing our study Determination of the prevalence of Microsatellite instability, BRAF and KRAS/NRAS mutation status in patients with colorectal cancer in Slovakia.

We are honoured to be guided by you as an experienced researcher in our publishing to the special issue focused on the Significance of KRAS Gene Mutations in Colorectal Cancer. Your valuable suggestions help with strengthening our manuscript and also to navigate us in our further research, as we are aiming to improve the understanding of the high morbidity and mortality of CRC patients in Slovakia and subsequently all patients suffering from this disease with increasing global incidence and its socioeconomic burden.

This retrospective article serves as a doorway to conduct further, more extended prospective research, focused on newly enrolled patients with NGS-based screening for Lynch syndrome-associated mutations, treatment, and subsequent follow-ups.

Thank you for your contribution We are looking forward to collaborating with you in our future submissions to Cancers. 

Please see the attachment of the word document, we follow up with replies to your comments and suggestions.

Kind regards

Research team

Reviewer 2 Report

Comments and Suggestions for Authors

1.        As the goal for this study is to determine the prevalence of microsatellite instability, BRAF and KRAS/NRAS mutations in colorectal cancer patients in Slovakia, the sample size of this cohort (83) is too small to represent the real prevalence. Could the authors increase the size of the cohort by including more patients before 2020?

2.        A pending question would be what factor(s) does make Slovakia as one of the leading countries with high incidence of colorectal cancer? Compared to the world’s distribution of MSI, BRAF and KRAS/NRAS mutations, is there any significant difference observed in Slovakia cohort?

Author Response

(The authors gave the same response as above.)

Reviewer 3 Report

Comments and Suggestions for Authors

The manuscript has been well-written. The author has conducted a descriptive analysis looking at the presence of biomarkers among confirmed CRC patients. For the study aim, the methods are appropriate. The tables correctly represent the data. The discussion section needs to provide a couple of points on the study strengths and limitations. The study is restricted to very few clinical factors. 

Line 44- replace the ',' with decimal point for 43,9 cases.

Line 211- Which cohort is being referred in this statement? Clarifying this would make it better as the numbers do not match to the above mentioned study numbers. 

Author Response

(The authors gave the same response as above.)

Reviewer 4 Report

Comments and Suggestions for Authors

Dear Authors,

Thank you for submitting your manuscript entitled "Determination of the prevalence of Microsatellite instability, BRAF and KRAS/NRAS mutation status in patients with colorectal cancer in Slovakia" for consideration. Your work addresses a significant gap in the current understanding of colorectal cancer (CRC) biomarkers within the Slovak population, which is of particular interest given the high incidence rates of CRC in the country. The analysis of microsatellite instability (MSI), BRAF, and KRAS/NRAS mutations and the evaluation of neoplastic cell percentage in tumor tissues present valuable insights that could influence future diagnostic and therapeutic strategies.

Research design, cohort specification, and methods are well described.

However, to strengthen your manuscript, I recommend the following things:

  1. Can you state the research aim and novelty in the introduction more clearly?
  2. Expand the discussion to compare your findings with international studies, highlighting unique aspects of CRC in Slovakia and the potential impact on research and clinical practices. This could offer a clearer view of how your results fit into the global understanding of CRC.
  3. Provide a deeper analysis of the rare co-occurrence of BRAF and KRAS mutations, discussing its implications for CRC pathogenesis and targeted therapy. A focused discussion on these dual-driver mutations' biological significance and potential treatment implications would be insightful.
  4. Elaborate on the study's limitations more comprehensively, especially the selection criteria's impact on the sample's representativeness. Discussing the challenges and biases introduced by the selection process and reliance on archived samples will provide a more complete understanding of the study's scope and accuracy.
  5. Please write references according to the Cancers instructions for authors.

Your contribution is valuable to the field, and with these enhancements, your study could offer a more robust resource for researchers, clinicians, and policymakers involved in CRC management in Slovakia and potentially other regions with similar CRC profiles.

Author Response

(The authors gave the same response as above.)

Round 2

Reviewer 2 Report

Comments and Suggestions for Authors

The authors addressed my concerns. Therefore, I would recommend it for publishing.